# The Effect of Extended Ball-Milling upon Three-Dimensional and Two-Dimensional Perovskite Crystals Properties

**Sara Bonomi [1,†], Vincenza Armenise [2,†] , Gianluca Accorsi [3] , Silvia Colella [4], Aurora Rizzo [3], Francesco Fracassi [2,4], Lorenzo Malavasi [1,*] and Andrea Listorti [2,3,*]**

1 Department of Chemistry and INSTM, University of Pavia, Via Taramelli 16, 27100 Pavia, Italy; sara.bonomi01@universitadipavia.it

2 Department of Chemistry, University of Bari 'Aldo Moro', via Orabona 4, 70125 Bari, Italy; vincenza.armenise@uniba.it (V.A.); francesco.fracassi@uniba.it (F.F.)

3 National Research Council, Institute of Nanotechnology (CNR-NANOTEC), c/o Campus Ecotekne, via Monteroni, 73100 Lecce, Italy; gianluca.accorsi@nanotec.cnr.it (G.A.); aurora.rizzo@nanotec.cnr.it (A.R.)

4 Department of Chemistry, National Research Council, Institute of Nanotechnology (CNR-NANOTEC), University of Bari 'Aldo Moro', via Orabona 4, 70125 Bari, Italy; silvia.colella@nanotec.cnr.it

\* Correspondence: lorenzo.malavasi@unipv.it (L.M.); andrea.listorti@uniba.it (A.L.); Tel.: +39-(0)382-987921 (L.M.); +39-080-5442009 (A.L.)

† S.B. and V.A. equally contributed to the work.

**Abstract:** The ball-milling of materials is a mechanical grinding method that has different effects on treated materials, and can be used for the direct synthesis of organometal halide perovskite (OHP) crystals. Herein, the effect of such a process, extended over a large temporal window, is related to the properties of referential three-dimensional (3D) MAPbI$_3$ (MA = methylammonium) and two-dimensional (2D) PEA$_2$PbI$_4$ (PEA = phenylethylammonium) perovskite crystals. For both 2D and 3D systems, the ball-milling induces a reduction of the crystallite dimension, accompanied by a worsening of the overall crystallinity, but without any sign of amorphization. For MAPbI$_3$, an intriguing room temperature structural transition, from tetragonal to cubic, is observed. The processing in both cases impacts on the morphology, with a reduction of the crystal shape quality connected to the particles' agglomeration tendency. All these effects translate to a "blue shift" of the absorption and emission features, suggesting the use of this technique to modulate the 3D and 2D OHPs' properties.

**Keywords:** ball-milling; organometal halide perovskites; crystals; 3D and 2D perovskite properties; optical properties

## 1. Introduction

In recent years, organometal halide perovskites (OHPs) have gained enormous attention as active materials applicable in several optoelectronic applications, such as solar cells, light emitting devices, photodetectors and field-effect transistors [1–6]. These materials indeed own exceptional optoelectronic properties, such as high absorption coefficients, the direct photogeneration of free carriers and long carrier diffusion length, and can be manufactured using numerous and low-energy consuming methods [1,7–10].

Most of the studies reported in the literature concern the investigation and application of thin film OHPs [9,11,12]. However, these films are usually characterized by a grain structure and morphology that directly impact on the optoelectronic features, since they are a collection of crystallites, amorphous phases and unreacted species [4,13–15]. Additionally, their optical and

electronic properties can be dominated by extrinsic defects, since trap states are more easily generated at the grain boundaries [11]. Therefore, in order to understand the intrinsic properties of these appealing materials, polycrystalline powdered samples represent an ideal model to be examined for the expected lower defect density (resulting from the absence of inter-grain regions), and for certain chemical compositions [5,8,12,13,16–19].

Peculiarly, the use of polycrystalline samples has nowadays gained significant attention with regards to the exploitation of the effects of different factors, such as chemical manipulation and external perturbations, on the structural, morphological and optoelectronic properties of the hybrid perovskites [20–34]. In some recent examples, polycrystalline samples were synthesized and investigated in order to (i) measure the impact of rubidium ions' incorporation into multiple-cation perovskites $Rb(MAFA)PbI_3$ (MA = methylammonium; FA = formamidinium) [21], (ii) study the role of different organic cations [27], (iii) study the effect of illumination [34] on $MAPbI_3$ properties, iv) explore the consequence of halide substitution in a lead-free $BZA_2SnX_4$ (BZA = benzylammonium; X = Cl, Br or I) system [28], (v) analyze the degradation processes triggered by the X-ray, $N_2$, $O_2$ or $H_2O$ exposure of $MAPbBr_3$ perovskite [29], and (vi) assess its enhanced photoluminescence and photoconductivity attributed to erbium-doping [22]. Perovskites crystals were also employed to evaluate the responses of OHPs to external temperature and pressure, which is considered a physical probe for tailoring the properties of these materials [25,33]. Indeed, the results of different temperatures and/or pressures on the structural and optoelectronic features of $MAPbI_3$ [22–25,32], $MAPbBr_3$ [30–32] and $FAPbI_3$ [33] crystals have been reported in the literature.

Herein, polycrystalline powdered samples of three-dimensional (3D) and two-dimensional (2D) OHPs are examined in order to evaluate the effects of prolonged high-energy ball-milling processes on their structural and optical properties. High-energy ball-milling is a mechanical grinding of materials, in which the milling occurs thanks to the mechanical friction between the material and a grinding medium (i.e., balls) included in a container [35–37]. This technique can have different consequences on the treated materials, such as different extents of downsizing, efficient mixing, solid-state chemical transformations, or a combination of all of these [36]. Very importantly, it can be applied as an environmentally friendly, reproducible, efficient and simple synthesis method for a variety of solid materials, among which are OHPs [24,35,38]. In particular, the use of this methodology should be considered as an alternative chemical strategy and a promising avenue for synthesizing high-purity OHP crystals in a solvent-free manner, which is expected to pave the way for the investigation of the fundamental properties (relevant to their implementation into optoelectronic devices) of these materials [35,38–42]. However, to our knowledge, few publications deal with the description of the process in detail, and in particular, with the study of its effects on the final OHPs samples. The present study aims to investigate the results of prolonged ball-milling on the structural, morphological and optical properties of $MAPbI_3$ 3D and $PEA_2PbI_4$ (PEA = phenylethylammonium) 2D perovskite crystals. For this purpose, polycrystalline samples were prepared via wet-chemistry methods, subjected to high-energy ball-milling for different times, and characterized by means of X-ray diffraction (XRD), scanning electron microscopy (SEM) and absorbance and photoluminescence (PL) spectroscopies.

## 2. Materials and Methods

### 2.1. Synthesis of Perovskites

The polycrystalline $MAPbI_3$ 3D and $PEA_2PbI_4$ 2D perovskite powders were synthesized using a wet-chemistry method [33] with the following steps: (i) dissolving stoichiometric lead (II) acetate (Sigma Aldrich; 99.9%) in a large excess (9 times for $MAPbI_3$, 15 times for $PEA_2PbI_4$) of hydriodic acid (57 wt % in $H_2O$, 99.99%, Sigma Aldrich) or hydrobromic acid (48 wt % in $H_2O$, ACS Reagent, Sigma Aldrich); (ii) heating up to 100 °C within an oil bath; (iii) adding a stoichiometric quantity of methylamine (40 wt % in $H_2O$, Sigma Aldrich) or phenethylamine (≥99%, Sigma Aldrich); (iv) letting

it cool down to 50 °C; (v) filtering by means of a water pump; and (vi) heating to 60 °C in a vacuum in a B-585 glass oven (BUCHI) for 10 h. All the first 4 synthesis steps were carried out in nitrogen flux.

### 2.2. High-Energy Ball-Milling Processes

Four grams of each synthesized powder were ground by means of a planetary ball-milling machine (Fritsch, Pulverisette 7, Premium Line) equipped with tungsten carbide (WC) jars and balls (each ball about 1 g) [28]. The samples were prepared with a powder–balls ratio of 1:10, by alternating 20 min of grinding at 800 rpm with a 10 min break, for total times of 240 and 420 h for $MAPbI_3$ and $PEA_2PbI_4$, respectively. The same amounts (~100 mg) of the $MAPbI_3$ samples were taken from each jar after 72, 120, 192 and 240 h; the same was done for $PEA_2PbI_4$ after 120, 204, 324 and 420 h.

### 2.3. Samples Characterization

XRD investigations of the powder samples were carried out with a Bruker D8 Advance diffractometer using copper Kα radiation (λ = 1.54056 Å) as the X-ray source, under an ambient condition (i.e., temperature 25 °C, humidity 25%). The measurements were performed in the Bragg–Brentano configuration, with a resolution of 0.02° and an integration time of 4 s.

A TESCAN Mira 3 high-resolution scanning electron microscope was used to perform the morphological characterization of the samples. SEM images were acquired at a working distance in the range 8.0–8.5 mm, an electron acceleration voltage of 20.00 kV, and 5.00–10.00 k× magnification.

Total absorption spectra were recorded with a PerkinElmer LAMBDA 1050 High Performance Series spectrophotometer including an integrating sphere for the collection of all the light scattered by the sample.

Steady state and time-resolved PL was measured by an Edinburgh FLS980 spectrometer equipped with a Peltier-cooled Hamamatsu R928 photomultiplier tube (185–850 nm). An Edinburgh Xe900 450 W Xenon arc lamp was used as the exciting light source. Corrected spectra were obtained via a calibration curve supplied with the instrument (lamp power in the steady state PL experiments ~0.6 mW/cm$^{-2}$, spot area 0.5 cm$^2$). Emission lifetimes were determined with the single photon counting technique by means of the same Edinburgh FLS980 spectrometer using a laser diode as the excitation source (1 MHz, $\lambda_{exc}$ = 635 nm, 67 ps pulse width and about 30 ps time resolution after deconvolution) and a Hamamatsu MCP R3809U-50 (time resolution 20 ps) as detector (laser power in the TRPL experiment ~1.6 mW/cm$^{-2}$, spot area 0.3 mm$^2$).

## 3. Results and Discussion

The powder XRD patterns of the $MAPbI_3$ 3D perovskite, after selected high-energy ball-milling times (0–240 h), are shown in Figure 1a. As can be seen, even after 240 h of the external treatment, the sample maintains a good crystallinity, with only a slight peak broadening with respect to the starting pattern, probably originating from strain-induced defects due to the mechanical treatment. The XRD data of the as-prepared (0 h of ball-milling) $MAPbI_3$ that overlapped with the peaks of the tetragonal crystal structure (vertical red bars) are reported in the bottom part of Figure 1b. As can be appreciated, all the expected peaks are present in the experimental pattern, confirming the typical tetragonal structure of the synthesized $MAPbI_3$ perovskite [25]. On the other hand, already considering the pattern of the sample after 72 h of milling (top part of Figure 1b), a clear phase transition occurs. The matching of the obtained diffraction peaks with those of the cubic phase (blue vertical bar) confirms the stabilization of this symmetry in the treated $MAPbI_3$ crystals. The clear sign of tetragonal symmetry [25,43], namely, the two peaks around 25°, are not present in the milled sample, suggesting that the high-energy mechanical treatment induces the structural phase transition of the 3D perovskite [32]. The cubic symmetry is also found in all the samples by further increasing the milling time, as displayed in Figure 1a. In addition, as reported in Table 1, the treatment has an effect on the lattice parameters, leading to a slight cell volume expansion as a function of time. The Le Bail approach was applied to calculate these parameters with a zero-shift value of −0.0651 [44].

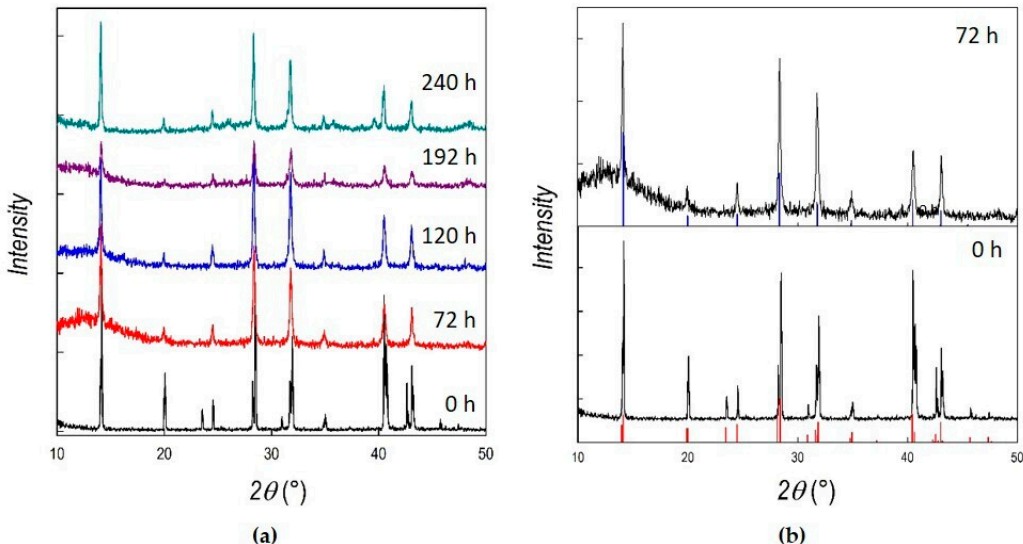

**Figure 1.** (**a**) X-ray diffraction (XRD) patterns of MAPbI$_3$ (MA = methylammonium) as a function of ball-milling time; (**b**) bottom: XRD pattern of as-prepared MAPbI$_3$ overlapped with the expected pattern for the tetragonal unit cell; top: XRD pattern of 72 h-milled MAPbI$_3$ overlapped with the expected pattern for the cubic unit cell.

**Table 1.** Lattice parameters and cell volume for MAPbI$_3$ at different ball-milling times.

| Crystalline Phase | Ball-Milling Time (h) | A (Å) | C (Å) | Cell Volume (Å$^3$) |
|---|---|---|---|---|
| Tetragonal | 0 | 8.868 ± 0.001 | 12.628 ± 0.001 | 248.300 |
| Cubic | 72 | 6.291 ± 0.001 | – | 248.977 |
| | 120 | 6.293 ± 0.001 | – | 249.214 |
| | 192 | 6.295 ± 0.001 | – | 249.452 |
| | 240 | 6.301 ± 0.001 | – | 250.166 |

The XRD patterns, as a function of the ball-milling time (0–420 h) for the PEA$_2$PbI$_4$ 2D perovskite, are displayed in Figure 2. The crystal structure of the as-prepared material is in agreement with the monoclinic space group, and the pattern is dominated by the 00l peaks, as is common in 2D metal halide perovskites [45,46]. A significant crystallinity reduction is found in the samples, already after 120 h, and it is progressively increased by raising the milling treatment time, as can be appreciated by the peak's broadening. In particular, after 420 h of ball-milling, most of the intensity is lost and the peaks are very broad. The FWHM moves from about 0.05° for the as-prepared sample, to about 0.49° after 420 h of milling time. At the same time, the first peak, corresponding to the 001 reflection, shifts to a lower angle by increasing the treatment, from 5.413° (0 h) to 5.238° (420 h), indicating an expansion of the c-axis. However, no sign of amorphization is found, suggesting the relevant robustness of the 2D sample structure. Furthermore, contrary to what was assessed for MAPbI$_3$, no evidences of phase transition are found in PEA$_2$PbI$_4$ perovskite crystals as a consequence of the ball-milling treatment at various times.

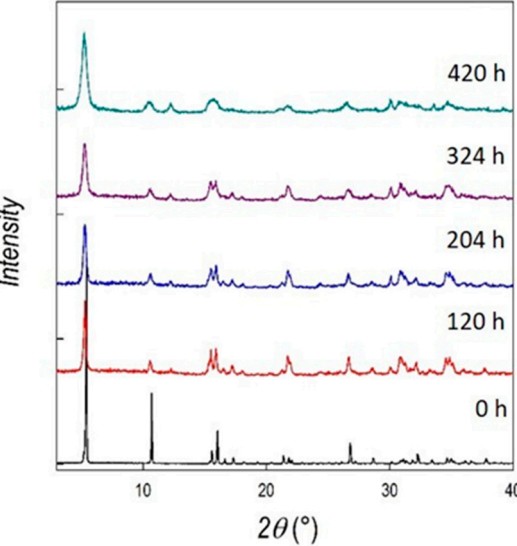

**Figure 2.** XRD patterns of $PEA_2PbI_4$ (PEA = phenylethylammonium) as a function of ball-milling time.

Representative SEM images of the as-prepared and after-ball-milling $MAPbI_3$ and $PEA_2PbI_4$ samples are respectively reported in Figures 3 and 4.

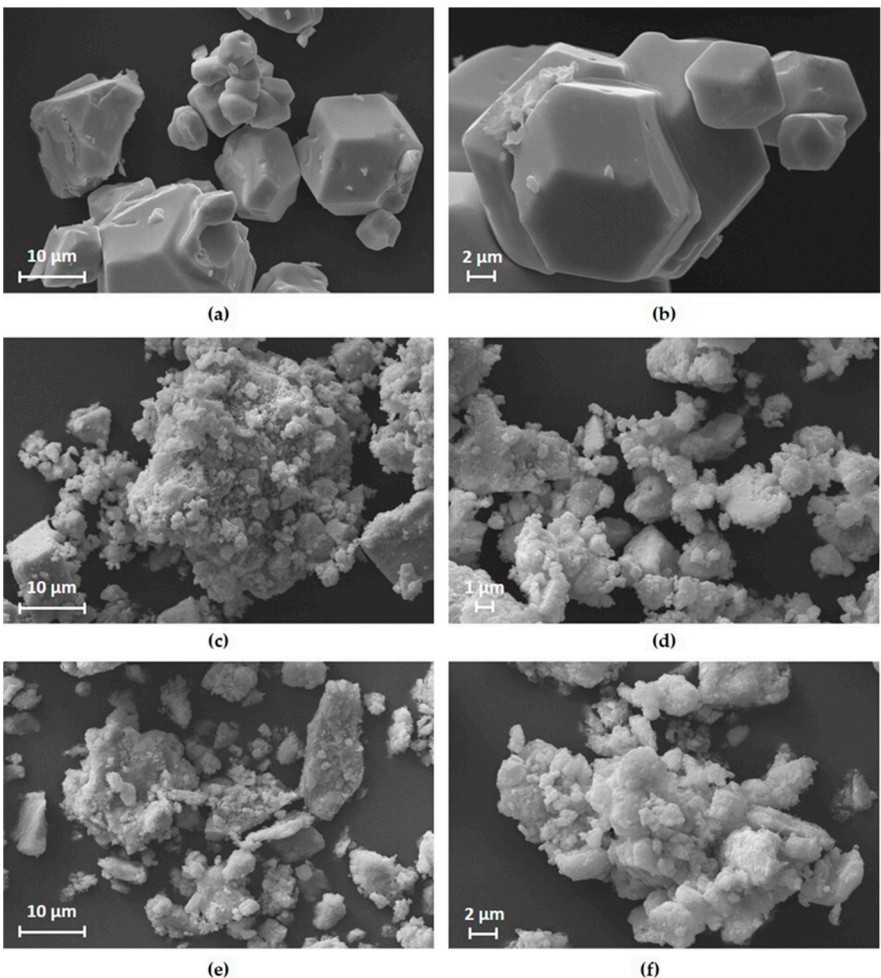

**Figure 3.** Scanning electron microscopy (SEM) images, at 5 k×and 10 k× magnification, of (**a**,**b**) as-prepared $MAPbI_3$, and that after (**c**,**d**) 120 and (**e**,**f**) 240 h of ball-milling.

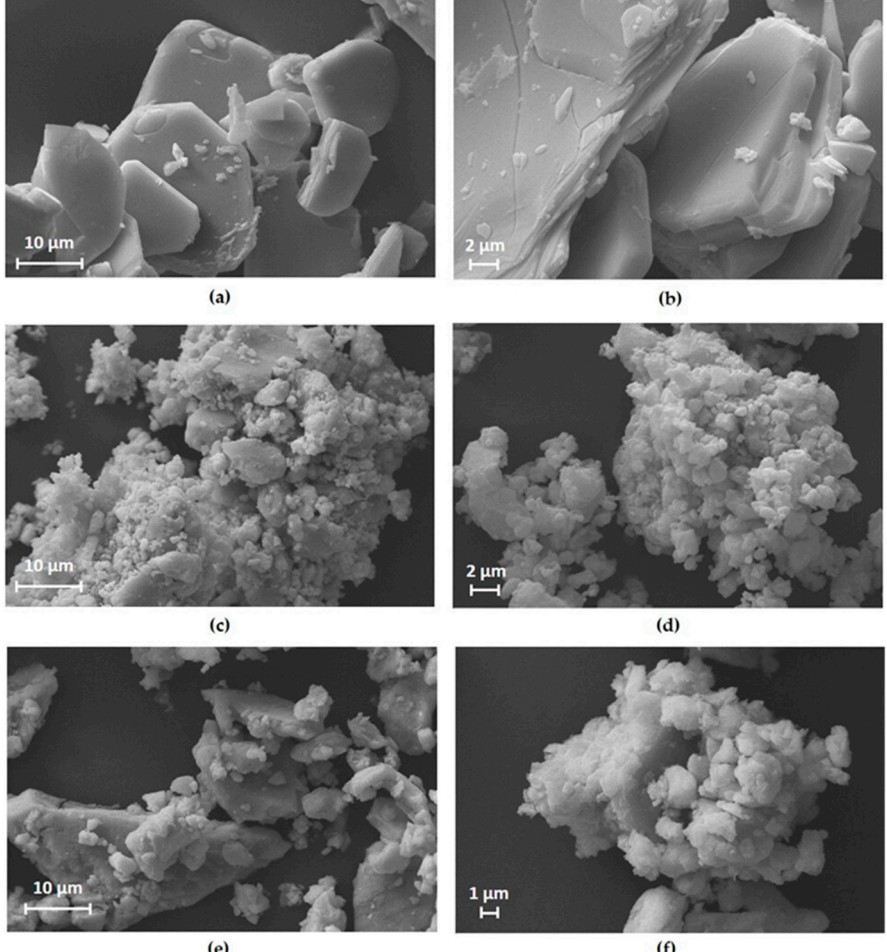

**Figure 4.** SEM images, at 5 k× and 10 k× magnification, of (**a,b**) as-prepared PEA$_2$PbI$_4$, and that after (**c,d**) 204 and (**e,f**) 420 h of ball-milling.

The as-prepared MAPbI$_3$ (Figure 3a,b) shows the typical cubic-like shape of the crystallites, which have dimensions ranging from 5 to 20–30 μm. The ball-milling is performed to significantly reduce the quality of the crystal shape, also producing smaller particles which tend to agglomerate (Figure 3c–f). No other remarkable variations concerning the morphology can be observed by increasing the milling time from 120 to 240 h.

As-prepared PEA$_2$PbI$_4$ (Figure 4a,b) reveals a lamellar-like morphology, with grains of several microns. Such a peculiar morphology is lost due to the ball-milling, which reduces the size of the crystallites, producing also a broader size distribution (Figure 4c,f). As in the case of MAPbI$_3$, by further prolonging the mechanical treatment (from 204 to 420 h), a dramatic change in the crystal morphology is not found. However, as is clear from the diffraction data, the intrinsic crystallinity is progressively reduced by increasing the ball-milling time, irrespective of the sample morphology and size.

For the 3D MAPbI$_3$ sample, the absorption spectra (Figure 5) show an evident shift associated with the ball-milling process. From the as-prepared sample to the 240-h-milled one, there is a "blue shift" of about 0.1 eV, with the sample after 120 h of treatment laying in an intermediate region. As discussed above, the high-energy process causes a phase transition from the tetragonal to the cubic perovskite phase. Additionally, it induces an expansion of the lattice. In principle, a reduction of the band gap, passing from a distorted Pb-I geometry (tetragonal) to a symmetric one (cubic), could be caused by an improved interaction between the lead and iodide orbitals. However, in the examined systems, the transition to the cubic phase is also linked to a lattice expansion, leading to an elongation of the

Pb−I bond length and thus to the downshift of the VB [47,48]. In the cubic phase "regime" of the MAPbI$_3$ perovskite, the [PbI$_6$]$^{4-}$ octahedra expansion (Table 1) increases the Pb−I bond length. This in turn reduces the coupling of the Pb s and the I p orbitals, and pushes down the VB, leading to the observed blue shift of the band gap. It is worth noting that the absorption edge retains a relative sharpness, even after long-lasting milling, which is an indirect indication of material order retention, as the low-energy tail of the exciton absorption is reminiscent of Urbach absorption, which is a measure of the disorder [49]. The emissions of the systems could not be obtained, for two main reasons; one is related to the setup used, which is very weakly sensitive above 830 nm, where the emission band likely lays [50], and the second is related to the very weak emission expected in such systems, due to the defects introduced following high-energy milling, as observed below for the more strongly emissive samples (PEA$_2$PbI$_4$).

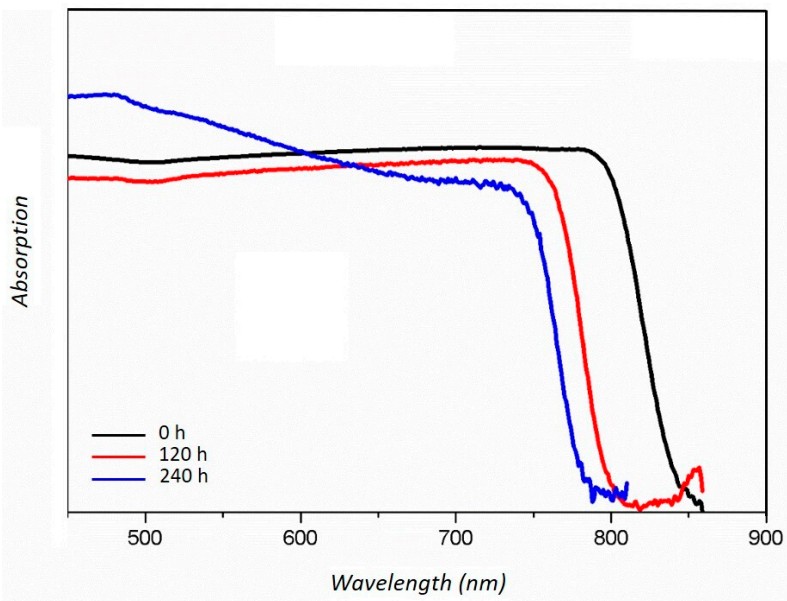

**Figure 5.** Absorption spectra of as-prepared (0 h) MAPbI$_3$, and after 120 and 240 h of ball-milling.

For PEA$_2$PbI$_4$ systems, a similar trend for the absorption spectra can be observed (Figure 6), also resulting in the light emission behavior, but with some additional findings. The as-prepared crystalline powder shows a narrow intense emission, characterized by a lifetime in the order of a few nanoseconds (biexponential decay: 2.5 and 10.5 ns), and the emission band presents a shift, relative to the absorption onset, of a few nm (around 8 nm). For the samples after 204 h and 420 h of high-energy treatment, consistent with the absorption features, a blue shift, a reduction in terms of the intensity of the emission compared to the referential compound, a smaller Stokes shift and a reduction of the emission lifetime can be noticed (for both samples the emission decays were below 50 ps). As clearly evidenced by the XRD measurements and the SEM images, the quality of the PEA$_2$PbI$_4$ crystals is strongly affected by the ball-milling, and the introduction of electronically active defects is logically linked to this process, impacting on the optical properties. However, of interest is the shift of the absorption and emission, which has been observed for 2D systems when quantum confinement effects were triggered by the system's reduced dimensionality. Herein, it is possible that some crystals, at least in one dimension, could shrink following ball-milling under the material Bohr radius, triggering some quantum confinement. Some of us recently found that, even in the bulk perovskite, a single photon emission could be triggered via material deposition protocol variation [51]; therefore, further studies will be dedicated to the issue.

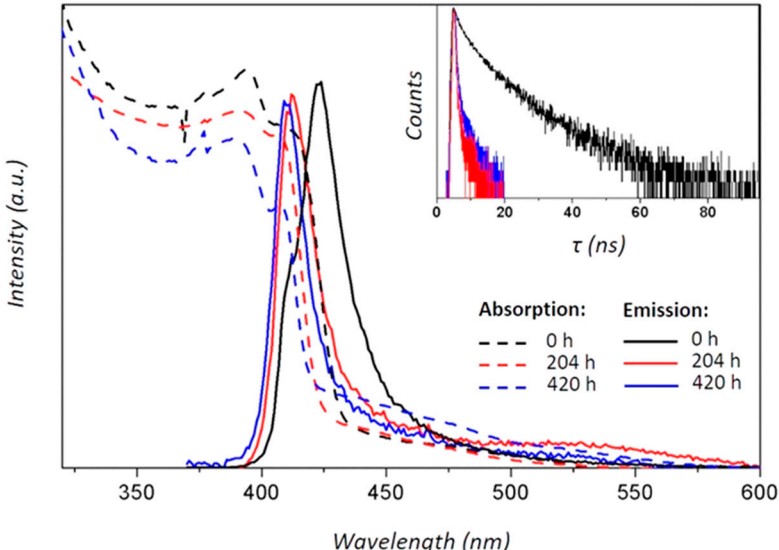

**Figure 6.** Absorption and steady-state emission photoluminescence (PL) spectra of as-prepared (0 h) $PEA_2PbI_4$, and that after 204 and 420 h of ball-milling (main panel). Time-resolved PL spectra (inset) of the same samples.

## 4. Conclusions

In this study, the effect of extended ball-milling on the structural, morphological and optical properties of $MAPbI_3$ 3D and $PEA_2PbI_4$ 2D perovskite crystals was investigated. Particularly, polycrystalline powders were synthesized via a wet-chemistry method; they then underwent high-energy ball-millings of different durations, and were analyzed using XRD, SEM, absorbance and PL spectroscopies. From the structural point of view, the XRD results showed no sign of amorphization for either sample, even after the prolonged milling treatment. Additionally, the processing induces a clear phase transition (from tetragonal to cubic) already after 72 h for $MAPbI_3$, and a significant crystallinity reduction for the $PEA_2PbI_4$ perovskite. The morphological SEM investigation displayed a reduction of the crystals' quality due to the ball-milling, with a reduction of their size and an agglomeration tendency. The optical characterization showed an interesting effect on the $MAPbI_3$ and $PEA_2PbI_4$'s band gap tunings, attributable to the high-energy treatment. Additionally, the 2D perovskite exhibited a reduction in terms of emission intensity and lifetime, probably due to the introduction of electronic defects, attributable the decreased quality of the crystals.

In conclusion, the mechanical ball-milling process affects many properties of crystalline $MAPbI_3$ 3D and $PEA_2PbI_4$ 2D perovskites, without incurring their amorphization, and it could be further applied to a wider class of OHPs in order to gain information regarding (micro)structure–properties correlations.

**Author Contributions:** Conceptualization, L.M., A.L.; methodology and investigation, S.B., V.A., S.C., G.A.; data curation and validation, L.M., A.L.; visualization, S.B., S.C., A.R., G.A.; writing—original draft preparation, V.A., A.L., L.M.; writing—review and editing, A.L., L.M., A.R., S.C., G.A.; supervision, resources and funding acquisition, L.M., F.F., A.R. All authors have read and agreed to the published version of the manuscript.

**Funding:** This research was funded by PON Project "Tecnologia per celle solari bifacciali ad alta Efficienza a 4 terminali per utility scale" (BEST-4U), of the Italian Ministry MIUR (CUP B88D19000160005)".

**Acknowledgments:** Alessandro Girella is acknowledged for SEM data collection.

**Conflicts of Interest:** The authors declare no conflict of interest.

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
