# Peer review of "The Effect of Extended Ball-Milling upon Three-Dimensional and Two-Dimensional Perovskite Crystals Properties"

_applsci, doi:10.3390/app10144775_

Round 1
Reviewer 1 Report
The manuscript titled ‘The effect of Extended Ball-Milling upon Three-Dimensional and Two-Dimensional Perovskite Crystals Properties’ submitted to Applied Sciences shows interesting results. This paper is well-organized and presented clear ideas, being adequate to be submitted. However, some minor clarifications ought to address before submitting it.
-Section 2.1: Synthesis of Perovskites. 6 methods or 6 sequent steps to synthetize the perovskite? Clarify this experimental preparation.
-The total time in the ball-milling machine of 240 h to MAPbI3 and 420h to PEA2PbI4. Why you choose these specific hours? Some references?
-How you calculate the lattice parameters, a and c, showed in Table 1? Add reference.
-XRD data in Figure 2. The authors should analyze the characteristic peak of 2D perovskites at 5 degree.
-Figure 5. Y-Axis should be corrected.
Author Response
Manuscript ID: applsci-835831
Title: The Effect of Extended Ball-Milling upon Three-Dimensional and Two-Dimensional Perovskite Crystals Properties
We would like to thank all the reviewers of the work, thanks to their punctual and careful evaluations/suggestions we were able to increase the soundness and the potential impact of the work.
The manuscript was amended accordingly and revisions were reported in red.
Response to Reviewer 1
- Section 2.1: Synthesis of Perovskites. 6 methods or 6 sequent steps to synthetize the perovskite? Clarify this experimental preparation.
Perovskite synthesis comprises six sequent steps. We revised the text accordingly in line 101
- The total time in the ball-milling machine of 240 h to MAPbI3 and 420h to PEA2PbI4. Why you choose these specific hours? Some references?
These times result from a series of preliminary experiments (not reported for the sake of brevity) and essentially both for the two materials represent total times where we could not see any further relevant change in the milled samples.
- How you calculate the lattice parameters, a and c, showed in Table 1? Add reference.
Lattice parameter were calculated with a Le Bail approach considering the relatively low-quality of the patterns due to ball milling, as added in line 153-154. This method is suited to extract the lattice parameters alone without further refinement of other parameters such as atomic positions or thermal factors. We added reference 44.
- XRD data in Figure 2. The authors should analyze the characteristic peak of 2D perovskites at 5 degree.
We analysed the peak position and FWHM of the first peak at about 5°. We found a progressive shift to lower angle by increasing the milling-time and a significant increase of the FWHM from about 0.05 to 0.492°. We added this part in the paper from line 177 to line 180.
- Figure 5. Y-Axis should be corrected.
We corrected Y-Axis.
Reviewer 2 Report
In this manuscript, Bonomi et al. investigate the effects of ball-milling presynthesized 2D and 3D hybrid organic-inorganic perovskites. While the topic may be of interest, there exist in my opinion several important misunderstandings in the data analysis. I therefore do not recommend the article for publication.
In detail, the authors study the effects of ball-milling MAPbI3 (3D) and PEA2PbI4 (2D).
On the 3D the main findings are a phase transition to cubic and a shrinkage of the unit cell. On the first point (phase transition), I agree that this is seen by XRD. To clarify their statement, it is the peak around 2theta=23.5º that is a clear sign of tetragonal phase ((121) plane; see https://doi.org/10.1021/acs.inorgchem.5b01481 ) and that vanishes upon milling. However, this is not surprising as this phase transition occurs at 54 ºC and 72 hours of milling will certainly get the powders to this temperature inside the jars. Also, the data is a bit noisy to be able to discern whether this peak is present or not in some cases (seems to be present in the red and turquoise diffractograms in Fig 1a). As a side-comment: why is the minimum time already so long (72 h) when it has been shown that much shorter times already have a significant impact (see DOI: 10.1039/C9TC03778K)?
On the second point, how did the authors determine the cell parameter of the cubic phase? I guess they must have done some Rietveld refinement or other type of whole-pattern fitting. How good are those fits (GoF, Chi-square, R-values…)? Is a variation lower than 0.01 A in the lattice parameter really significant here?
On the 2D material, it is claimed that crystallinity is reduced but no sign of amorphization is present. Could the authors clarify this statement? Isn’t a “reduction in crystallinity” the same as amorphization? What would the signs of amorphization be? Also, in Figure 2, the black diffractogram goes all the way to the top of the figure. This should be scaled down as the others. I guess then the minor peaks would not me appreciable, but this is meaningfull information. This probably means that somehow the anisotropy or preferential orientation on the substrate changes with ball-milling. To be discussed.
Other comments:
- “relatively long lifetime in the order of few nanoseconds”: that sounds rather short.
- “the so called stoke shift” should be Stokes shift. Also, it is not “narrower” as it does not have a width; it should be simply “smaller”.
Author Response
Manuscript ID: applsci-835831
Title: The Effect of Extended Ball-Milling upon Three-Dimensional and Two-Dimensional Perovskite Crystals Properties
We would like to thank all the reviewers of the work, thanks to their punctual and careful evaluations/suggestions we were able to increase the soundness and the potential impact of the work.
The manuscript was amended accordingly and revisions were reported in red.
Response to Reviewer 2
- On the 3D the main findings are a phase transition to cubic and a shrinkage of the unit cell. On the first point (phase transition), I agree that this is seen by XRD. To clarify their statement, it is the peak around 2theta=23.5º that is a clear sign of tetragonal phase ((121) plane; see https://doi.org/10.1021/acs.inorgchem.5b01481 ) and that vanishes upon milling. However, this is not surprising as this phase transition occurs at 54 ºC and 72 hours of milling will certainly get the powders to this temperature inside the jars. Also, the data is a bit noisy to be able to discern whether this peak is present or not in some cases (seems to be present in the red and turquoise diffractograms in Fig 1a). As a side-comment: why is the minimum time already so long (72 h) when it has been shown that much shorter times already have a significant impact (see DOI: 10.1039/C9TC03778K)?
We agree with the reviewer as for sure the samples reach a temperature where the tetragonal-to-cubic phase transition occurs. However, and this is quite interesting, this phase transition is metastable, since the samples are measured after opening the jars in the next hours at room temperature. In principle, by lowering the temperature the MAPI should return to the tetragonal phase transition. We collected experiments for many time intervals (also very short) but this paper aims at looking at the effect of prolonged milling, so data at low-time are not included. Finally, the data are not of optimal quality, but to our judgment of a quality level to rule out the presence of tetragonal phase.
- On the second point, how did the authors determine the cell parameter of the cubic phase? I guess they must have done some Rietveld refinement or other type of whole-pattern fitting. How good are those fits (GoF, Chi-square, R-values…)? Is a variation lower than 0.01 A in the lattice parameter really significant here?
We determined the lattice parameters by Le Bail method, as added in line 153-154. 0.01 A difference is significant. GoF parameters are within good values expected by whole-pattern fitting with Rwp around 10.
- On the 2D material, it is claimed that crystallinity is reduced but no sign of amorphization is present. Could the authors clarify this statement? Isn’t a “reduction in crystallinity” the same as amorphization? What would the signs of amorphization be? Also, in Figure 2, the black diffractogram goes all the way to the top of the figure. This should be scaled down as the others. I guess then the minor peaks would not me appreciable, but this is meaningfull information. This probably means that somehow the anisotropy or preferential orientation on the substrate changes with ball-milling. To be discussed.
Amorphization should be present in the pattern as a diffuse background. In no one of the data this feature is clearly evident, thus ruling our significant amorphization. We rescaled the black pattern in Figure. This sample is highly crystalline and with a morphology (see SEM images) which significantly enhance the (00l) diffraction. It is clear that preferential orientation changes as a consequence of the change of morphology of 2D material.
- Other comments:
- “relatively long lifetime in the order of few nanoseconds”: that sounds rather short.
We modified our comment
- “the so called stoke shift” should be Stokes shift. Also, it is not “narrower” as it does not have a width; it should be simply “smaller”.
We modified the sentence accordingly to the suggestion.
Reviewer 3 Report
I expressed my comments in the attached document.

Author Response
Manuscript ID: applsci-835831
Title: The Effect of Extended Ball-Milling upon Three-Dimensional and Two-Dimensional Perovskite Crystals Properties
We would like to thank all the reviewers of the work, thanks to their punctual and careful evaluations/suggestions we were able to increase the soundness and the potential impact of the work.
The manuscript was amended accordingly, and revisions were reported in red.
Response to Reviewer 3
General comments:
- In the introduction, the relevance of the ball-mill technique, the state-of-the-art status and the relevance of this work are not good enough explained.
We modified the introduction section accordingly to the reviewer comment (from line 85 to 92).
- The effect of phase transition is interesting, however the results presented do not add new understanding. The research question apparently was: Does the ball milling process and its duration affect the structural, morphological, and optical properties of 3-D MAPbI3 and 2-D PEA2PbI4 powders? Why only these two materials were chosen? PEA2PbI4 is not strictly saying a perovskite. Did authors observe a reverse phase transition in MAPbI3 after some time? How the milling influences phase stability implying decomposition in PbI2 and organic counterpart?
We selected the two referential representative for 2D and 3D organometal halide perovskites (OHPs), these two materials as cornerstone referential for the optoelectronic exploitation of OHPs. The phase transition is very interesting metastable in our report, since the samples are measured after opening the jars in the next hours at room temperature. In principle, by lowering the temperature the MAPbI3 should return to the tetragonal phase transition, but it does not following this specific treatment. As far as for the detection limits of our XRD measurements, we could not observe any evidence of materials decomposition.
- Palazon et. al / ( Mater. Chem. C, 2019, 7, 11406) synthesised CsPbBr3 by ball milling and analysed the effect of process duration. A quote from their abstract: “at much longer milling times (up to 10 hours) eventually smaller quantum-confined CsPbBr3 NCs are exfoliated from the bulk product leading to a broad and blue shifted emission. At this stage, the photoluminescence intensity is strongly reduced which is ascribed to the formation of surface defects induced by ball-milling in dry conditions”.
We thank the reviewer for the suggestion and we inserted the reference in the paper (number 42).
- The blue shift in extended milling of 2-D perovskite leads to blue shift in Eg. This seems to be a well know “Quantum Confinement” phenomenon where change in physical dimensions (formation of 2-D platelet for example) leads to a change in optical properties from bulk. (see: ACS Sustainable Chem. Eng. 2018, 6, 3, 3733–3738, ACS Nano 2016, 10, 7830−7839 and Growth Des. 2017, 17, 794−799)
We thanks the reviewer for the suggestion and those interesting references. For a different project we found also quantum effects in perovskite materials depending on materials deposition procedures. We modified the text accordingly from line 241 to line 248 and added reference 51.
- The absence of PL intensity for MAPbI3 can be explained by the increase non-radiative trap sites induced by ball-milling? Instead of the pointed low sensitivity of device? The PL emission is expected around 790 nm. ( Mater. Chem. C, 2019, 7, 11406, J. Mater. Chem. A, 2015, 3, 20772–20777 )
We agree with the reviewer for the trap induced emission reduction, so we modified the text from line 221 to line 226. Regarding the emission band expectation the one of large crystals can fall above 830 nm (reference 50).
- Line 124: Cubic transformation of MAPbI3: This point is interesting. Is this ball milling pressure induced (J. Phys. Chem. C 2019, 123, 30221−30227)? or effect of negative surface tension on nano crystal (ACS Energy Lett. 2020, 5, 238−247)?
It is hard to properly define the origin of phase transition in MAPbI3, also because HP studies are generally performed at high pressure ranges. The reference (J. Phys. Chem. C 2019, 123, 30221−30227) indicated by the reviewer is particularly interesting because it evidences a tetragonal to cubic phase transition at particular low-pressure which can be though reasonable for a ball-milling process. We added this reference in the related part of the discussion.
- There is no clear explanation, why milling results in significantly smaller crystallites in case of PEA2PbI4 (compare FWHMs in the Fig. 2 ) but not in case of MAPbI3 (Fig. 1)? We strongly recommend to perform the microstructural XRD analysis extended with the Williamson-Hall plots (see: J. Mater. Chem. C, 2019, 7, 11406) to obtain the apparent average crystal sizes and discuss these results.
We thanks the reviewer for this suggestion. In general for both samples there is a reduction of crystal size (as we better discussed in the revised manuscript) and a loss of the original morphology. Due to the nature of the milling process it is difficult to define a proper size distribution in the milled samples which is very broad, this indicates that the application of Williamson-Hall plot is not particularly suited for this specific case to define an apparent average crystal size. In addition differently to what obtained in “J. Mater. Chem. C, 2019, 7, 11406”, in our case the extended milling (much longer time) lead to a severe reduction of XRD spectra quality in particular for PEA2PbI4 this again suggest that Williamson-Hall plot is not particularly suited for this specific case. We believe that the most relevant information to pass to the reader is contained in the morphological evidence resulting from SEM analysis.
Style, spelling, small remarks:
- In general, there is an abuse of using the parentheses; this interrupts the continuity of the text and makes reading difficult.
We modified the draft trying to avoid when possible to interrupt its readability.
- Passive voice is used with too long introductory sentences.
We modified the introduction accordingly to reviewer suggestion
- Line 55: (MAFA)PbI3 this not the composition. From ref [21] (FAPbI3)9(MAPbBr3)0.05(CsPbBr3)0.05
We controlled and corrected the material composition indication.
- Line 53-60: Sentence needs clarification.
We modified the paragraph accordingly to reviewer suggestion.
- Line 81: no reference is provided for the synthesis of perovskites.
We added a reference.
- Line 86: “within” instead of "with".
We modified the word.
- Line 86-87: "(40 wt.% in H2O, Sigma Aldrich) or phenethylamine (≥99%, Sigma Aldrich)" - what does this "or" depends on?
The use of methylamine (40 wt.% in H2O, Sigma Aldrich) or phenethylamine (≥99%, Sigma Aldrich) depends on the composition of the perovskite. Indeed, we used methylamine and phenethylamine to synthesize MAPbI3 and PEA2PbI4, respectively.
- Line 87: "Cool up" is a wrong expression.
We modified our expression.
- Line 89: "for one night" - how many hours is this?
We modified the text: 10 hours.
- Line 95: "the same amount" – not specified.
An aliquot of about 100 mg. We added this information in the text.
- Line 99: "under ambient condition" – specify at least temperature and relative humidity.
Temperature (25 °C) and humidity (25%) were specified.
- 1: Which model was used for the refinement, Rietveld, Le Bail, other one? Why the same analysis was not performed for the PEA2PbI4 (Fig.2)? Assignment of the observed diffraction maxima would be informative.
We used the Le Bail method (reference 44). We included in the manuscript the analysis of the diffraction characteristic of PEA2PbI4 here the broadening of the peaks did not encourage additionally studies.
- 2: peaks ~70 and ~110 overlap in a bad way. Potentially interesting effects of the relative intensity change get lost/not discussed.
We modified the Figure.
- Line 153: "cubic-like" - however it has been established that this structure is tetragonal.
This comment concerns the shape of the crystallites that appears similar to cubes.
- Line 156: "A not remarkable change" - please define.
We defined it in line 191.
- Line 172: "blue jump" – sounds like a slang.
We modified the sentence.
- Line 175: "then" - the following sentence do not provide a consequence of the previous one.
We modified the sentence.
- Line 190: "this translates" - use "resulting" instead.
We modified the text.
Reviewer 4 Report
This paper presents results of ball-milling on the structural and optical properties of 2D and 3D organometal halide perovskite. The paper is organized well and has can be consider for publication but after minor revision.
Line 60 ….. “erbium-doping [26]” must be checked again.
Line 120, The XRD shows broadening of the peak, is that because of the smaller crystals? It can be also due to presence of some defects. We can see the broadening of the peak located at ~32°. A short discussion is required.
Line 178, author claimed about the lattice expansion. That is fine, but the discussion about the lattice expansion is missing in XRD part! For instance, the peak located between 28-29° is shifted slightly after ball-milling.
The legends of Figure 1a and Figure 2 are not clear. My suggestion is to reverse their orders according to corresponding spectra. Or alternatively, write down name of each spectra on their right side.
In addition, it would be much clear to denote the main peaks in the Figure 1 and 2.
Author Response
Manuscript ID: applsci-835831
Title: The Effect of Extended Ball-Milling upon Three-Dimensional and Two-Dimensional Perovskite Crystals Properties
We would like to thank all the reviewers of the work, thanks to their punctual and careful evaluations/suggestions we were able to increase the soundness and the potential impact of the work.
The manuscript was amended accordingly, and revisions were reported in red.
Response to Reviewer 4
- Line 60 ….. “erbium-doping [26]” must be checked again.
There was a mistake related to the number of the reference. We modified it.
- Line 120, The XRD shows broadening of the peak, is that because of the smaller crystals? It can be also due to presence of some defects. We can see the broadening of the peak located at ~32°. A short discussion is required.
We believe the broadening is mainly due to strain effects induced by high energy ball milling. We added a sentence in the paper in lines 141.
- Line 178, author claimed about the lattice expansion. That is fine, but the discussion about the lattice expansion is missing in XRD part! For instance, the peak located between 28-29° is shifted slightly after ball-milling.
In the revised version of the manuscript X-Ray diffraction discussion has been expanded, we believe this new version replies to this remark.
- The legends of Figure 1a and Figure 2 are not clear. My suggestion is to reverse their orders according to corresponding spectra. Or alternatively, write down name of each spectra on their right side.
We modified the figures writing the ball-milling time of each spectra on their right side.
- In addition, it would be much clear to denote the main peaks in the Figure 1 and 2.
Since the two materials are benchmark perovskites for optoelectronic, we preferred to leave the figure without superimposed writings.
Round 2
Reviewer 2 Report
The authors have not really given any further evidence to reply to my original comments, which I will repeat and possibly clarify hereafter:
- The XRD data in Figure 1 is not sufficient to rule out the presence of tetragonal MAPI in the ball-milled samples. If anything, the characteristic peak around 23.5º seems to be present after 72h of ball-milling. Ideally the authors should acquire diffractograms with better signal-to-noise ratio to discuss this point. If further experiments or measurements are not possible, at least a "zoom" in the region of interest (around 2theta=23.5º) should be presented.
- Concerning the lattice parameters, it would be beneficial if the authors include the details of the Le Bail fits (lattice parameter values with error, zero-shift... etc.).
Author Response
Manuscript ID: applsci-835831
Title: The Effect of Extended Ball-Milling upon Three-Dimensional and Two-Dimensional Perovskite Crystals Properties
Response to Reviewer 2
- The XRD data in Figure 1 is not sufficient to rule out the presence of tetragonal MAPI in the ball-milled samples. If anything, the characteristic peak around 23.5º seems to be present after 72h of ball-milling. Ideally the authors should acquire diffractograms with better signal-to-noise ratio to discuss this point. If further experiments or measurements are not possible, at least a "zoom" in the region of interest (around 2theta=23.5º) should be presented.
As suggested by the reviewer to better address her/his concern, we present a zoom of Figure 1 in the region of interest. As can be observed the characteristic peak around 23.5° is not present after 72 h of ball-milling as well as for longer ball-milling times, confirming the absence of tetragonal MAPbI3 in the ball-milled samples.
Figure. XRD patterns of MAPbI3 as a function of ball-milling time
- Concerning the lattice parameters, it would be beneficial if the authors include the details of the Le Bail fits (lattice parameter values with error, zero-shift... etc.).
We included the value of the zero-shift and the lattice parameter values with error in the revised manuscript in line 140 and Table 1 (highlighted in yellow).

Reviewer 3 Report
Please improve text quality further, e.g.:
line 63: "exposure f MAPbBr3" - "of" sample perhaps?
line 79: " ball-milling can be defined a type of" - is to rephrase;
line 201: " as clear from the diffraction data" - no verb;
line 213: "due to a possible improved interaction" - sounds odd;
line 237: "Stoke shift" should be displaced by "Stokes shift".
Author Response
We thank the reviewer for her/his suggestions and we modified the paper accordingly.
Round 3
Reviewer 2 Report
The authors have provided further information on the obtained values and error and a "zoomed" version of XRD data to better assess the presence or absence of peaks.
While I still believe that better signal-to-noise ratios would be welcome for the analysis of XRD data, this may not be considered a compulsory requisite for publication.
Author Response
We thank the reviewer for her/his appreciation of our work